# GENOMIC FOUNDATIONLESS MODELS: PRETRAINING DOES NOT PROMISE PERFORMANCE

## ABSTRACT

The success of Large Language Models has inspired the development of Genomic Foundation Models (GFMs) through similar pretraining techniques. However, the relationship between pretraining performance and effectiveness in downstream genomic tasks remains unclear. Additionally, the high computational cost of pretraining raises questions about its cost-efficiency. To assess the usefulness of pretraining in genomics, we evaluated seven different GFMs across various benchmarks, comparing them to their counterparts with randomly initialized weights. Surprisingly, we found that randomly initialized models can match or even surpass the performance of pretrained GFMs in finetuning and feature extraction tasks. We also discovered that pretrained GFMs fail to capture clinically relevant genetic mutations, which are crucial for understanding genetic disorders and phenotypic traits. Our results indicate that most of the current pretrained GFMs lack a "foundational" understanding of genomics and provide minimal utility, even for basic tasks such as sequence classification. These findings collectively highlight the need for critically rethinking the pretraining approaches for genomics. Our code is available at github.com/nxifemwt/GFMs.

## 1 INTRODUCTION

Recent advances in language modeling have led to the application of similar unsupervised pretraining approaches in genomics. This facilitated the emergence of Genomic Foundation Models (GFMs) (Consens et al., 2023) which learn representations from genomic sequences. This line of work has attracted considerable attention due to the potential of GFMs to revolutionize our understanding of genomics (Benegas et al., 2024).

GFMs typically use a two-step training approach akin to Large Language Models: unsupervised pretraining on a large dataset, followed by a supervised training. The pretraining phase usually involves either next token prediction (Brown et al., 2020) or masked language modeling (Devlin et al., 2018). The promise of unsupervised pretraining is to extract knowledge from vast genomic datasets (Consortium et al., 2015) and compress it into the model's parameters, with the aim of producing a generalist model applicable to a diverse set of tasks.

While some studies have explored scaling laws for GFMs (Nguyen et al., 2023; 2024), the relationship between pretraining and downstream performance remains unclear, with no single GFM consistently proving to be the best (Marin et al., 2023). Combined with large model sizes (Dalla-Torre et al., 2023), long input sequences (Nguyen et al., 2023; 2024) and massive datasets, the pretraining step demands substantial computational resources. The natural question arises: *how effective is unsupervised pretraining in the genomics domain?*

To answer this, we conduct extensive experiments with seven recent GFMs across multiple benchmarks, as shown on Figure 1, comparing the performance of pretrained models to their randomly initialized counterparts. Our study reveals that randomly initialized models often perform competitively with pretrained models or even surpass them, suggesting that current pretraining approaches may not provide a significant advantage over random weight initialization. Specifically, we found that on the Nucleotide Transformer Benchmark (Dalla-Torre et al., 2023), GUE (Zhou et al., 2024), and Genomics Benchmark (Schiff et al., 2024) randomly initialized models trained from scratch in a supervised manner perform either better then or on par with finetuned pretrained GFMs.

Figure 1: **Overview of the experiments.** (A) **Finetuning:** Experiments are performed on NT Benchmark where we finetune models for functional element classification tasks. (B) **Feature Extraction:** For biotype classification, we extract embeddings from frozen models and train a simple classifier to predict gene types using these embeddings. (C) **Genomic Variation:** We evaluate models' ability to capture genetic variations through two tasks: (1) Mutation sensitivity analysis measures how well models distinguish between original and mutated sequences by computing embedding similarities, and (2) Ancestry prediction uses model embeddings with XGBoost to classify population groups based on genomic variants. Both tasks use sequences constructed by combining HG38 reference genome with mutation data.

| Model | #Params | Architecture | Tokenizer | Vocab Size | Seq Len (tokens) | #Tokens | Data |
|-------|---------|--------------|-----------|------------|------------------|---------|------|
| HyenaDNA | 450K | Decoder | Char | 12 | 1024 | 2.6B | HRG |
| NT 500M | 500M | Encoder | k-mer | 4107 | 1000 | 300B | 1000G |
| NTv2 50M | 50M | Encoder | k-mer | 4107 | 2048 | 300B | Multispecies |
| GENA-LM | 110M | Encoder | BPE | 32000 | 512 | 1T | HRG+1000G |
| DNABERTv2 | 117M | Encoder | BPE | 4096 | 128 | 262B | Multispecies |
| Caduceus | 8M | Decoder | Char | 12 | 131K | 35B | HRG |
| Mistral | 580M | Decoder | Char | 12 | 4096 | 150B | 1000G |

Table 1: **Description of models evaluated in this study.** The analyzed models differ in architecture, pretraining objective, tokenizer, model size, and pretraining dataset. We analyze the pretrained models and their randomly initialized counterparts. *#Tokens* refers to the number of tokens seen by the model during the pretraining. *Data* refers to the pretraining dataset source.

In addition to finetuning, we examine feature extraction tasks to assess the quality of the representations learned during the pretraining. These tasks involve extracting features from the model with frozen weights and applying a simple classifier to these embeddings. For randomly initialized models, this means that the weights remain as originally randomly initialized, without any tuning at all. Intuitively, one would expect untrained models with randomly initialized weights to perform poorly compared to pretrained.

One such task is biotype classification, where the goal is to predict the functional type of a genomic sequence. Surprisingly, randomly initialized models demonstrate competitive performance compared to pretrained in biotype classification. Moreover, simple modifications, such as changing the tokenizer and increasing the embedding dimension, significantly boost the performance of randomly initialized models, enabling the completely untrained HyenaDNA to outperform all pretrained GFMs on this benchmark.

Another important group of tasks uses mutation information as the primary predictive feature. These scenarios require models to be highly sensitive to single nucleotide changes within long sequences. We found that most pretrained GFMs fail in these tasks. For instance, even when up to half of the nucleotides in a DNA sequence are changed, some GFMs still produce embeddings with over 0.99 cosine similarity to the original sequence. As a result, GFMs are currently unsuitable for applications that rely extensively on mutation data, including variant pathogenicity prediction, eQTL (Zhou & Troyanskaya, 2015), sQTL (Garrido-Martín et al., 2021), and phenotype prediction.

Overall, our results challenge current unsupervised pretraining methods used in genomics, suggesting that simply adapting NLP techniques is insufficient for developing true genomic understanding.

Rather than continuing to invest substantial computational resources in existing pretraining methods, we advocate for critically rethinking the fundamental building blocks of genomic foundation models. This includes developing biologically-informed tokenization strategies and establishing new robust benchmarks that comprehensively test for the understanding of genomic mechanisms.

## 2 MODELS

We selected six recently published GFMs for evaluation and also trained our own version of the Mistral (Jiang et al., 2023) model on 50 samples from the 1000 Genomes dataset (Consortium et al., 2015). The models in our analysis exhibit significant diversity in their architectures, pretraining objectives, tokenizers, model sizes, and pretraining datasets. Our model selection includes both encoder and decoder architectures, transformer-based and state-space models, with model sizes ranging from 450K to 500M parameters. Interestingly, our Mistral outperforms all other previous GFMs on many tasks. We attribute the success of Mistral to an advanced architecture recipe which includes RoPE embeddings, big embedding dimension and character tokenizer. Model configurations are summarized in Table 1, and model descriptions are provided in Section A.1 of the Appendix.

We excluded the EVO model (Nguyen et al., 2024) from our analysis as it was trained on bacterial genomes and performed poorly in our preliminary tests on the Nucleotide Transformer Benchmark.

**Random weight initialization** of models throughout the paper follows the procedure from the Transformers library (Wolf et al., 2020) for each particular model. This usually involves initializing linear layers with values drawn from $\mathcal{N}(0, 0.02)$, and LayerNorm layers are initialized with $\gamma = 1$. Full random initialization details for each model are provided in Section A.2 in Appendix.

## 3 EXPERIMENTS

### 3.1 FINETUNING

To verify the usefulness of pretraining, we finetuned both pretrained and randomly initialized versions of the models on **Nucleotide Transformer Benchmark** (Dalla-Torre et al., 2023), **Genome Understanding Evaluation (GUE)** (Zhou et al., 2024), and **Genomic Benchmarks** (Grešová et al., 2023) with exactly the same set of hyperparameters. This set of benchmarks together constitutes **52** genomic classification tasks. In total, we conducted nearly **10,000** finetuning experiments, this considers: seven models, both pretrained and random, evaluated across different tasks, folds, and learning rates. Full hyperparameter details for these experiments are provided in Section A.4 in the Appendix. We display our results for these finetuning experiments in Figure 2.

For each task, we first find the highest score among all randomly initialized models; for example, if the scores obtained from randomly initialized models are 0.3, 0.4, and 0.5, we consider 0.5 as the best random score. We then plot the difference between each pretrained model's performance and the best random score. Green bars show where pretrained models outperform the best random model, while red bars show where they underperform. Ideally, if the pretraining is useful, we expect to see a predominance of tall green bars.

NT Benchmark results for histone and enhancer tasks are displayed in the top part of Figure 2. For the GUE Benchmark, we aggregate results by task categories. For example, in the Epigenetic Marks category, we average the scores across all histone modification tasks. Similarly, we compute average scores for other categories: Promoter Detection, TF Prediction Human and Mouse, Core Promoter Detection, and Splice Site Detection (which contains a single task). These aggregated results are presented in the middle part of Figure 2. For Genomic Benchmarks we display the performance for six different tasks in the bottom part of Figure 2.

**The results in Figure 2 demonstrate that big pretrained models often perform worse than small randomly initialized models.** This is visible by the big proportion of the red bars indicating that the best random model performance is higher than of the pretrained models. Notably, the randomly initialized Caduceus, despite having only 8M parameters, emerges as the best random model in six out of twelve tasks on NT Benchmark, four out of six tasks on GUE, and in two out of six tasks on Genomic Benchmarks. In general, randomly initialized Caduceus significantly outperforms larger pretrained models, including NT 500M, NTv2 50M, GENA-LM with 110M parameters, and DNABERTv2 with 117M parameters, and often even its own pretrained version. On NT Benchmark tasks of H3K9ac, H3K4me1, and H3K36me3 the randomly initialized Caduceus outperforms NTv2

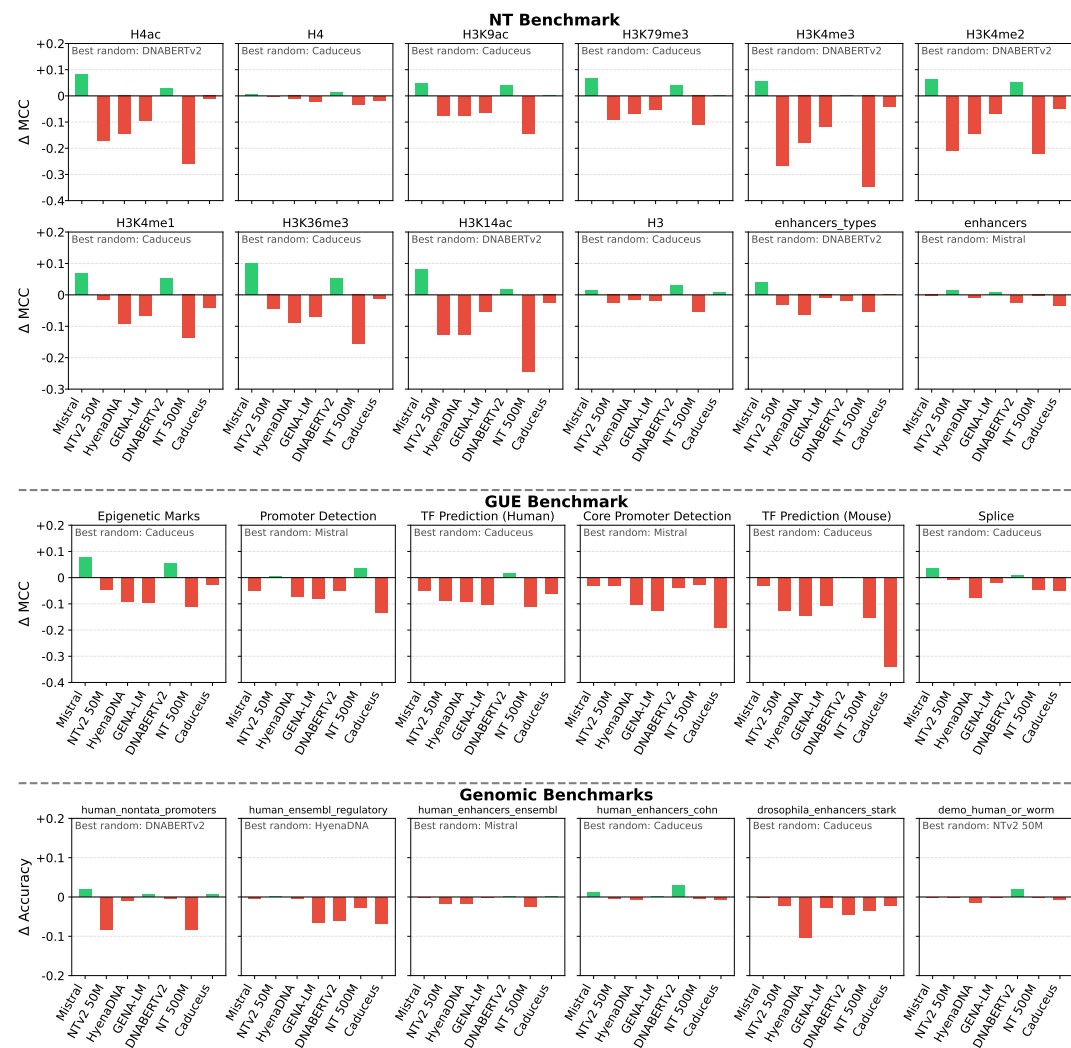

Figure 2: **Difference of performance between pretrained and the best random model on NT Benchmark.** For each task, we finetuned each model, starting from both pretrained and randomly initialized weights. Green bars indicate the advantage of pretrained models, and red bars indicate the advantage of the best random model. The best random model consistently outperforms several pretrained ones on each task, highlighting the inefficiency of current pretraining approaches in genomics. In most cases, the best random model is Caduceus which has only 8M parameters, yet it has better performance than much bigger pretrained models such as NT 500M, GENA-LM, DNABERTv2, NTv2 50M, and Mistral.

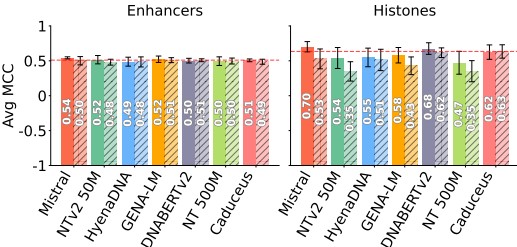

Figure 3: **NT Benchmark performance per subgroup.** Pretrained models are shown with clear bars, and randomly initialized with dashed. For enhancer subgroup all random models show competitive performance with pretrained. For histones random Caduceus outperforms five pretrained model including its own pretrained version. Red dashed line indicates MCC score of the best randomly initialized model.

50M, HyenaDNA, and GENA-LM, it is also better than NT 500M by about 0.1 MCC, **while being 60x times smaller (8M vs 500M)**.

Another good randomly initialized model is DNABERTv2. On challenging histone tasks on NT Benchmark, the randomly initialized DNABERTv2 with 117M parameters outperformed the pretrained NT 500M by about 0.35 MCC on H3K4me3 and by more than 0.2 MCC on H3K4me2 and

H3K14ac. This difference in MCC is quite significant. In general, for most datasets, the best randomly initialized model outperformed, on average, three or four pretrained models and consistently achieved performance comparable to the best pretrained model.

The results on GUE in the middle part of Figure 2 demonstrate an even more pronounced advantage of randomly initialized models compared to the NT Benchmark. In TF Prediction (Mouse), randomly initialized Caduceus shows remarkable performance, outperforming all pretrained models. Similar trend is observed on Core Promoter Detection Group where randomly initialized Mistral outperforms all pretrained models including its own pretrained version. In general on GUE benchmark the best randomly initialized model outperforms five to seven pretrained models.

In Genomic Benchmarks similar trend of competitiveness of randomly initialized models can be observed. For example, randomly initialized HyenaDNA that only has 450K parameters outperforms all pretrained models on human_ensembl_regulatory task and the same is true for randomly initialized Caduceus on drosophilia_enhancers_stark task.

To showcase the individual performance of randomly initialized models, we present their results on NT Benchmark alongside pretrained models in Figure 3. For instance, the "Enhancers" subgroup includes all enhancer-related tasks, while the "Histone" subgroup covers all histone tasks, and so on. In addition, we also show this plot for Splice Sites and Promoter on NT Benchmark in Figure 7, and also for GUE and Genomic Benchmarks in Figure 8 and Figure 9 in the Appendix. We also provide results for all models on NT Benchmark in Table 15 in Appendix.

The results presented in Figure 3 highlight that randomly initialized models can perform remarkably well across all subgroups of the NT Benchmark. In the "Enhancers" subgroup, all randomly initialized models perform comparably to their pretrained counterparts. In histone tasks, the best random models, DNABERTv2 and Caduceus, reach average MCC scores of 0.62 and 0.63, outperforming pretrained NT 50M, HyenaDNA, GENA-LM, and NT 500M. In case of randomly initialized Caduceus it also outperforms its own pretrained version.

The results across all three benchmarks demonstrate that while not all randomly initialized models consistently outperform pretrained ones, we identified several randomly initialized models like Caduceus, DNABERTv2, HyenaDNA that can match or exceed pretrained performance across a wide range of tasks. Moreover, even in cases where pretrained models maintain an advantage, the gains from pretraining are surprisingly small - typically within 2-3%.

> ***Finding 1:*** Randomly initialized models can perform competitively with, and even surpass, pretrained models in finetuning tasks. Notably, this competitiveness is not a function of model size - the randomly initialized Caduceus with only 8M parameters consistently outperforms much larger pretrained models including NT 500M, GENA-LM (110M parameters), DNABERTv2 (117M parameters), and NTv2 50M. This pattern holds robustly across three different genomic benchmarks and types of tasks.

**Pretraining gains in genomics vs in other domains.** The success of foundation models in computer vision and NLP has been built on clear substantial gains from unsupervised pretraining. For example, CLIP (Radford et al., 2021) showed 10-30% improvements in robustness to distribution shift compared to standard ImageNet models, while GPT (Brown et al., 2020) few-shot performance outperformed SOTA finetuned model in question-answering tasks. However, our experiments show a different pattern in genomics, where the best randomly initialized models often outperform pretrained ones, and in case when pretrained models are better the difference with randomly initialized is generally marginal, i.e. about 2-3%. together those gains do not justify large amounts of compute needed for pretraining in genomics (Dalla-Torre et al., 2023).

> ***Finding 2:*** While pretraining provides double-digit improvements in computer vision and NLP, the gains in genomics are typically within 2-3% and often negative, challenging the effectiveness of current genomic pretraining approaches.

## 3.2 Feature Extraction

The biotype classification task assesses the quality of features extracted from the models. On this benchmark, we also compared the performance of both pretrained and randomly initialized models. However, unlike in the NT Benchmark where models were finetuned, in this task, we did not modify

the model weights at all. This means that *embeddings for randomly initialized models were extracted without any finetuning and were entirely based on their initial random weights.*

| Tokenizer | Pretrain | Decoder-only | | | Encoder-only | | | |
| | | Mistral | HyenaDNA | Caduceus | NTv2 50M | GENA-LM | DNABERTv2 | NT 500M |
|---|---|---|---|---|---|---|---|---|
| default | ✓ | 0.730 | 0.638 | 0.423 | 0.679 | 0.704 | 0.654 | 0.662 |
| default | ✗ | 0.667 | 0.690 | 0.674 | 0.482 | 0.574 | 0.651 | 0.603 |
| char | ✗ | 0.666 | 0.690 | 0.674 | 0.642 | 0.668 | 0.696 | 0.669 |
| +larger embed dim | ✗ | 0.700 | 0.753 | 0.717 | 0.703 | 0.684 | 0.708 | 0.678 |
| pretrained − random | | **3.0%** | **-11.5%** | **-29.4%** | **-2.4%** | **2.0%** | **-5.4%** | **-1.6%** |

Table 2: **Biotype classification results.** Embeddings extracted from pretrained and randomly initialized models were used to train an XGBoost classifier. Switching to character tokenizer (3rd row) and increasing the embedding dimension (4th row) significantly improved performance, allowing most randomly initialized models to surpass their pretrained counterparts. The bottom row shows difference in performance between pretrained model and optimized randomly initialized model. Negative values indicate the advantage of the random models. F1 score is reported.

Using sequences and biotype labels from the Gencode repository (Harrow et al., 2012), we extracted features from models with frozen weights and applied max pooling along the token dimension. These pooled features were then used to train an XGBoost classifier to predict among nine biotype labels. Detailed information about the dataset is presented in Section A.5 of the Appendix.

We observed that the choice of tokenizer significantly impacts the performance of randomly initialized encoder-only models. In particular, switching these models from their default k-mer or BPE tokenizers with large vocabularies (Table 1) to a character tokenizer that has only four tokens substantially improved their performance (third row of Table 2, right part). For example, for NT 50M, the performance increased from 0.48 to 0.64. Character-level tokenization is standard for decoder models, hence the identical results in the second and third rows of Table 2 for decoder-only models.

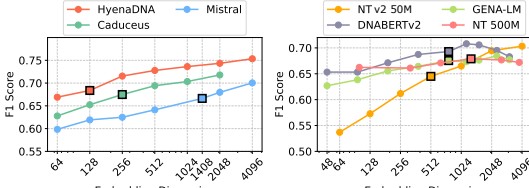

Figure 4: **Embedding dimension experiments.** We change embed_dim in randomly initialized models when using together with char tokenizer. The default embed_dim values are shown with square marker. Increasing embed_dim improves the performance.

The improvement after switching the tokenizer likely occurs because random models struggle with the large vocabulary size of the default tokenizer (Table 1). In contrast, the char tokenizer reduces the model's search space to just four tokens, making it easier for random models to make predictions.

Initially, the random HyenaDNA model stood out as the best among all random models, achieving an F1 score of 0.69 despite using a relatively small embedding dimension of 128. This observation prompted an investigation into the impact of increasing the embedding size on performance. We conducted a comprehensive sweep of embedding dimensions for all selected models, keeping all other parameters constant. It is important to note that the embedding dimension had to be divisible by the number of attention heads, which varied among models, necessitating different embedding dimensions for each model. Full configurations for embedding values used for each model are provided in Table 8 in the Appendix.

Figure 4 presents detailed plots for the embedding dimension experiments. It reveals a clear trend of improved performance as the embedding dimension increases for all five models examined. HyenaDNA shows consistent improvements, reaching an F1 score of nearly 0.75 at 4096 dimensions. NT 50M exhibited a more dramatic improvement, with its F1 score rising from 0.53 to 0.71. Additionally, we performed the same set of experiments on 10 Histone modification tasks from GUE benchmark. As shown in 10, random HyenaDNA with embedding dimension of 2048 is best on 9 out of 10 tasks, outperforming every pretrained model.

As shown in the fourth row of Table 2, increasing the embedding dimensions and using a character tokenizer allowed randomly initialized models to outperform pretrained in 5 out of 7 instances.

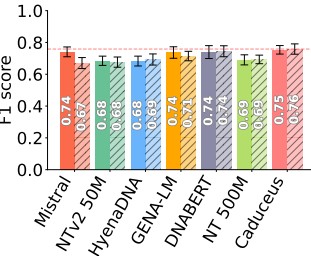

Figure 5: **Ancestry classification performance.** Pretrained models are indicated with clear bars, and randomly initialized counterparts are shown with dashed bars. F1 score is averaged over eleven regions. Randomly initialized Caduceus is achieving the highest score outperforming all pretrained models.

> *Finding 3:* For biotype classification, embeddings from pretrained models do not show a clear advantage over those from models with random weights. Additionally, random models optimized with simple changes, like swapping the tokenizer and increasing the embedding dimension, outperform pretrained models.

### 3.3 GENOMIC VARIATION

In this section, we shift our focus from functional element classification tasks to genomic variation tasks, which represents a fundamentally different problem in genomic analysis. Genomic variation tasks focus on the mutations in DNA sequences between individuals. These mutations, including single nucleotide polymorphisms (SNPs), insertions, and deletions, are crucial for understanding human genetic diversity and its implications for personal health.

Unlike functional elements, which are largely consistent across individuals, genomic variations are unique to individuals or populations and can significantly affect phenotype and disease risk. These variant-based analyses use mutation information as the primary predictive feature, necessitating models to detect and interpret subtle sequence differences between individuals.

This group of tasks presents a critical challenge for GFMs: they must be highly sensitive to small variations, as the genetic differences between populations often come down to single nucleotide polymorphisms (SNPs) scattered throughout the sequence. This requirement tests the models' ability to detect and interpret subtle genetic variation effects.

#### 3.3.1 ANCESTRY PREDICTION

Ancestry prediction is a multilabel classification task that involves predicting an individual's ancestry using only a short portion of their genome. In our experiments, we first constructed an ancestry dataset using the 1000G data (Consortium et al., 2015). We used the HG38 and applied mutations from every 1000G sample (Figure 1 C) to obtain consensus sequences for each individual. We fixed the sequence length to 32K bases. These sequences differ by approximately 0.5% of positions and, on average, contain 33 variants, including SNPs, insertions, and deletions. For each sequence, we generated embeddings which served as features for XGBoost classification.

When generating the dataset, we selected eleven different regions of the genome, treating each as a separate fold, and evaluated our models on each region independently, reporting average metrics (Figure 5). A detailed description of the benchmark is available in Section A.6 in the Appendix.

As shown in Figure 5, randomly initialized models can match the performance of pretrained models in ancestry prediction. Only the Mistral and GENA-LM pretrained models perform slightly better than their randomly initialized counterparts, with an F1 difference of 0.02. The best overall model is Caduceus, achieving an F1 score of 0.71 for both its random and pretrained versions. Notably, even the NT 500M model, specifically trained on data containing variants from 1000G, fails to outperform its randomly initialized version.

This can be attributed to the combination of two factors: the masked language modeling objective with a high masking probability (15%) and the k-mer tokenization strategy. The masking ratio far exceeds the natural mutation rate (0.5%), while the k-mer tokenizer, which processes sequences in chunks of 6 nucleotides, is poorly suited for capturing single nucleotide variations. Together, these design choices make it difficult for the model to learn meaningful representations of genetic variants.

#### 3.3.2 MUTATION SENSITIVITY ANALYSIS

To further investigate the reasons for poor model performance in ancestry prediction (Section 3.3.1) and to assess the capability of pretrained models in capturing relevant genomic variations, we conducted experiments to evaluate their sensitivity to mutations in DNA sequences. These experiments

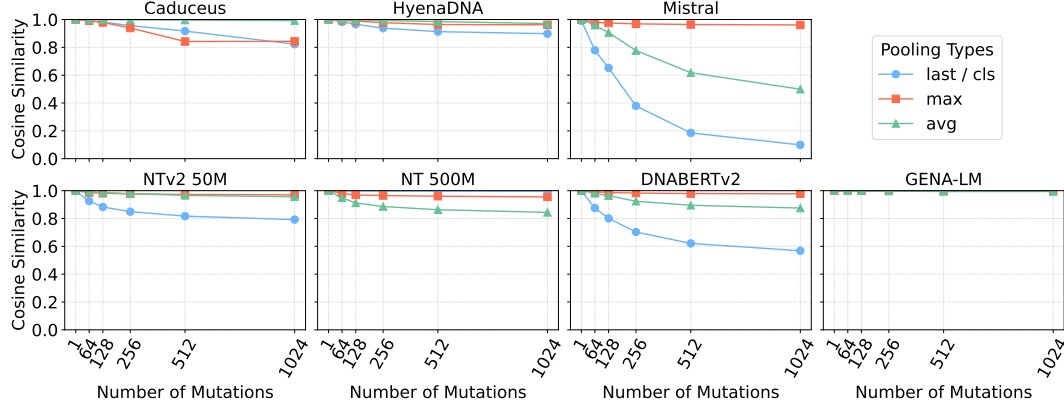

Figure 6: **Mutation sensitivity plot.** Although cosine similarity decreases with more mutations, the values remain high, indicating the models are mostly insensitive to the mutations. For blue markers, depending on the model type (encoder or decoder), we use either last or cls tokens.

aimed to evaluate the models' ability to detect differences between reference sequences and sequences with inserted variants.

To isolate the effect of genetic variations and eliminate sequence length as a confounding factor, we focused solely on single nucleotide polymorphisms (SNPs). By maintaining consistent sequence lengths across all mutated sequences, we ensure that any differences in embeddings were attributable to the SNPs themselves. We measure the cosine similarity between the embeddings of the original DNA sequences and their modified counterparts.

High cosine similarity scores on this test would suggest that the model treats altered sequences as nearly identical to the reference, overlooking biologically significant changes. A highly sensitive model would exhibit lower cosine similarity between reference and altered sequences, indicating its ability to differentiate them.

To conduct this experiment, we first sample 1024-length sequences from specific locations in HG38, targeting chromosomes 7, 11, 12, 17, and 19. In total, we sample 25 sequences, extracting 5 sequences from different regions of each targeted chromosome. We choose a length of 1024 because it fits within every model's context window without chunking, eliminating possible chunking effects from the analysis. In each sequence, we introduced mutations at gradually increasing levels (1, 64, 128, 256, 512, 1024) at random, unique positions without replacement. Embeddings for both the reference and mutated sequences are then generated using different pooling methods: last/cls tokens (based on the decoder or encoder architecture), max pooling, and average pooling.

Figure 6 illustrates the cosine similarity between reference and altered sequences across different pooling types. Despite the models using different tokenizers, the results are generally poor across the board. For most models, average and max pooling produced consistently high cosine similarities ($> 0.9$) even for a large number of mutations. Last or cls token embeddings tended to produce lower cosine similarity scores for DNABERTv2, NT 50M, and Mistral.

Our Mistral model stood out as having the lowest cosine similarity among all models for last pooling and relatively low cosine similarity for average pooling. In contrast, GENA-LM produced high cosine similarity scores close to 0.999 for all pooling types. These results indicate that most models are not significantly affected by mutations, thereby highlighting their limited ability to detect subtle sequence alterations, irrespective of their tokenization strategies. Similar trends are observed for randomly initialized models where most models produce high cosine similarity scores except NTv2 50M. Results for randomly initialized models are presented in Figure 10.

### 3.3.3 CLINVAR EXPERIMENTS

To further investigate the sensitivity of genomic models to sequence alterations, we conducted additional experiments using ClinVar data (Landrum et al., 2014), which includes genetic variations among individuals. These experiments aim to verify our previous findings in a more realistic setting, utilizing real-world genetic variations from ClinVar. We chose to analyze the TP53, BRCA2 and CFTR genes and obtained their gene sequences from the NCBI database (Sayers et al., 2022).

| | | NT 500M | NTv2 50M | DNABERTv2 | HyenaDNA | Mistral | GENA-LM | Caduceus |
|---|---|---|---|---|---|---|---|---|
| TP53 | Benign | 0.985 | 0.991 | 0.995 | 0.999 | 0.976 | 1.000 | 0.985 |
| | Pathogenic | 0.983 | 0.993 | 0.996 | 0.999 | 0.988 | 1.000 | 0.990 |
| BRCA2 | Benign | 0.999 | 0.984 | 0.964 | 0.996 | 0.907 | 0.996 | 0.996 |
| | Pathogenic | 1.000 | 0.984 | 0.955 | 0.999 | 0.981 | 1.000 | 0.973 |
| CFTR | Benign | 1.000 | 0.998 | 0.998 | 1.000 | 0.999 | 1.000 | 0.999 |
| | Pathogenic | 1.000 | 0.999 | 0.998 | 1.000 | 0.996 | 1.000 | 0.999 |

Table 3: **Gene-specific Variant Detection Performance**. Average performance metrics across different models for TP53, BRCA2, and CFTR genes, showing benign and pathogenic variant detection capabilities. Lower values indicate better performance in distinguishing variants.

First, we filtered the variants to include only exonic mutations. This step ensures a focus on mutations that affect protein-coding regions, which are often of greatest interest in clinical genetics. Next, we categorized the variants into two groups based on clinical significance: benign and pathogenic. The benign group included variants labeled as 'Benign', 'Likely benign', or 'Benign/Likely benign', while the pathogenic group comprised variants classified as 'Pathogenic', 'Likely pathogenic', or 'Pathogenic/Likely pathogenic'. This grouping enables us to compare the model's sensitivity to mutations with different clinical impacts.

After preprocessing the data, we take five chunks of 1024 base pairs for each gene independently that have both benign and pathogenic mutations. For each chunk, we created three versions: a reference sequence without mutations, a sequence with only pathogenic mutations, and a sequence with only benign mutations. The distribution of mutations is shown in Table 13 in Appendix.

This variation in mutation density allows us to observe the model's sensitivity across different levels of sequence alteration. For each chunk, we applied max pooling to the model outputs and computed the cosine similarity between the reference sequence and both the benign and pathogenic versions, repeating this process for each model. Finally, we averaged cosine similarity over five selected chunks. The results presented in Table 3 showed consistently high similarity scores across all models and mutation types, regardless of the number of mutations in each chunk, indicating the consistent failure of models to reflect genomic variance in their embeddings.

> *Finding 4:* Current pretrained GFMs exhibit poor performance on variant-based tasks, which can be attributed to their lack of sensitivity to sequence mutations.

## 4 RELATED WORKS

**Genomic Foundation Models.** Encoder-only approaches have proven effective in sequence prediction tasks, using k-mer tokenization (Ji et al., 2021; Dalla-Torre et al., 2023), Byte Pair Encoding (Zhou et al., 2024; Sanabria et al., 2023), and learnable vector quantization codebooks (Li et al., 2024) to enhance efficiency and manage longer sequences. Certain encoder architectures have been enhanced with recurrent memory mechanisms (Fishman et al., 2023) to capture long-range dependencies more effectively, while others utilize whole-genome alignments (Benegas et al., 2023) to incorporate evolutionary context. More recent work has explored pan-genome graph representations (Zhang et al., 2024) to better capture genetic variation diversity.

Meanwhile, decoder-only architectures have shown potential by integrating structured state-space models (Nguyen et al., 2023; Schiff et al., 2024), achieving competitive performance with minimal parameters and supporting long context lengths. Hybrid architectures (Nguyen et al., 2024), incorporating both attention and state-space blocks, have emerged, demonstrating great generative capabilities spanning from molecular to genome scales. Our work introduces a GFM based on Mistral architecture (Jiang et al., 2023) and performs performance analysis of the most recent GFMs.

**Genomic Foundation Models Analysis.** It was shown that k-mer embeddings pretrained on random DNA sequences can reach similar performance to those of trained on the real-world biological data (Zhang et al., 2023). Another study found that character tokenization outperforms other methods in state-space models (Lindsey et al., 2024). Evaluation of GFMs across the BEND benchmark reveals that they capture limited information on long-range features (Marin et al., 2023). It was also shown that mean pooling improves performance of GFMs for genomic sequence classifications and closes the performance gap between them (Feng et al., 2024). Pretrained DNA models were benchmarked (Tang et al., 2024) showing they do not offer great advantage over conventional machine

learning methods. In contrast to this study, our analysis includes finetuning and variant-based tasks, more models and also shows that randomly initialized models can be better as feature extractors.

# 5 DISCUSSION

## 5.1 INEFFECTIVENESS OF CURRENT GFM PRETRAINING METHODS

GFMs pretrained on vast amounts of data on many GPUs promise to revolutionize our understanding of genomics. However, our study reveals a surprising reality: despite their name, current GFMs lack substantial "foundation" in genomic understanding. Across multiple benchmarks, we consistently found that randomly initialized models perform competitively and sometimes outperform, their pretrained counterparts. The competitive performance of randomly initialized models suggests that current pretraining methods for GFMs are ineffective.

This finding is particularly significant given the substantial computational resources and associated costs required for model pretraining, often involving weeks of processing on many high-performance GPUs (Dalla-Torre et al., 2023). Instead, the pretraining process seems to function merely as a sophisticated, yet resource-intensive, weight initialization technique.

We also assessed the performance of GFMs on one of the most critical set of tasks in genomics which try to understand the impact of mutations in various applications. Many tasks in this domain require processing long sequences while maintaining sensitivity to single nucleotide changes. Our experiments in Section 3.3 demonstrate insensitivity of current GFMs to these crucial genetic variations. This limitation persists regardless of the tokenization and pooling methods used.

It is important to clarify that we do not claim that current pretraining methods for GFMs are without any merit at all. They still may offer advantage in specific contexts, such as generative tasks (Nguyen et al., 2024). However, our research shows that the current pretraining approaches do not yet realize the potential of creating truly generalist models with broad applicability. The hallmark of foundation models, as demonstrated by GPT and BERT in NLP, is their ability to generalize across a wide spectrum of applications. Our results show that there is room for improvement in genomic pretraining strategies to better capture this essence of versatility.

## 5.2 ALTERNATIVE APPROACH TO FOUNDATION MODELS

Collectively our findings indicate that simply scaling up the pretraining further is unlikely to yield significant benefits in genomic understanding. An alternative approach would be to focus on solving fundamental problems with specialized architectures. One such example is AlphaFold's breakthrough in protein structure prediction (Jumper et al., 2021). Rather than aiming to be a foundational model with broad scope of application, AlphaFold targeted a specific yet fundamental challenge in structural biology. Its success can be attributed to the development of a protein-specific architecture, the availability of high-quality data (Berman et al., 2007), and the existence of the universally respected CASP competition, which provided a clear metric for optimization (Moult et al., 2020).

Importantly, while AlphaFold focused on a specific problem, its success had far-reaching implications. It drew significant attention to the field and catalyzed numerous other impactful projects. For instance, AlphaMissense (Cheng et al., 2023), a state-of-the-art method for predicting protein variant pathogenicity, was built upon AlphaFold. This illustrates how solving a core problem can naturally lead to a model becoming foundational through its wide-ranging applications.

# 6 CONCLUSION

We conducted a comprehensive evaluation of Genomic Foundation Models by comparing pretrained models with their randomly initialized counterparts across multiple benchmarks. Our experiments reveal that randomly initialized models can perform competitively with, and sometimes surpass, their pretrained versions in both finetuning and feature extraction tasks.

Additionally, our analysis of genomic variation tasks demonstrated that current GFMs lack sensitivity to mutations, limiting their utility in many clinical applications. These findings challenge the effectiveness of current pretraining approaches in genomics and suggest that the substantial computational resources invested in pretraining may not yield proportional benefits. We hope our analysis will encourage the development of more effective approaches for creating genomic models that truly capture the complexities of biological sequences.

## REPRODUCIBILITY STATEMENT

The datasets for Biotype Classification and Ancestry Prediction will be made available upon release. The data for NT Benchmark is publicly available here. Additionally, dataset statistics and the details of train-test split are provided in the Appendix for each task. For each task, training configuration and hyperparameters have been provided in the Appendix section. Code for some of our experiments is provided in github.com/nxifemwt/GFMs, we will upload more code upon full release. For NT Benchmarks, we reported test metrics using a 3-fold cross-validation approach. For ancestry experiments we averaged the results over eleven different folds. For all experiments requiring a human genomic reference, we consistently use the GRCh38.p14 assembly (also known as hg38.p14).

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

# A APPENDIX

## A.1 MODELS

- **HyenaDNA** (Nguyen et al., 2023): Decoder-only state-space model with 450K parameters. Uses character tokenizer and was pretrained on the Human Reference Genome with a 1024 base pair sequence length.

- **Caduceus** (Schiff et al., 2024): Decoder-only model with 8M parameters. Trained on sequences of 131k base pairs on HRG. Combines a bidirectionally equivariant decoder with character tokenizer.

- **Mistral** (our version): Decoder-only transformer model with 500M parameters. Uses character tokenization and was trained on the 1000 Genomes dataset (Consortium et al., 2015). Details for pretraining are provided in Section A.3 in Appendix.

- **Nucleotide Transformer** (Dalla-Torre et al., 2023): Encoder-only model presented in two versions: a 500M parameter model trained on the 1000 Genomes Project data and its v2 with 50M parameter model trained on multispecies data. Both use k-mer tokenization.

- **GENA-LM** (Fishman et al., 2023): Encoder-only model with 110M parameters. Employs BPE tokenizer and was pretrained on the HRG with 1000G augmentations.

- **DNABERTv2** (Zhou et al., 2024): Encoder-only model with 117M parameters. Uses BPE tokenization and was trained on multispecies data.

## A.2 RANDOM WEIGHT INITIALIZATION

We initialized the model weights following a procedure using standard Hugging Face Transformers library initialization methods:

- For linear layers: Weights were initialized from a normal distribution $\mathcal{N}(0, 0.02)$, biases were initialized to zero.

- For LayerNorm: The scaling factor (gamma) was initialized to 1. The bias term (beta) was initialized to 0.

- For Embedding Layers: Embeddings were initialized from the same normal distribution $\mathcal{N}(0, 0.02)$.

  For Caduceus and HyenaDNA we performed prenorm residual rescaling, which is the default weight initialization procedure for these models. Biases for linear layers were initialized as zeros.

## A.3 MISTRAL PRETRAINING

We pretrain a Mistral model on 50 random individual samples from the Genome1000 project. Table 4 provides the Mistral configuration details and Table 5 provides the Mistral training configuration. Specifically, reverse complement of sequences formed with Genome1000 VCFs is used with a probability of 0.5. All the chromosomes (chr1 - chrX) are used for sequence formation from 50 individuals. Individuals are sampled in a stratified way, 10 from each superpopulation. We filtered out sequences where number of unknown nucleotides was more than half of sequence length. Total number of tokens is 150B.

| config | value |
|---|---|
| num_hidden_layers | 16 |
| num_attention_heads | 16 |
| hidden_size | 1408 |
| vocab_size | 12 |
| intermediate_size | 7168 |

Table 4: **Mistral model architecture.**

| config | value |
|---|---|
| tokenizer | character |
| sequence_len | 4096 |
| num_epochs | 1 |
| initial_lr | 7.2e-4 |
| final_lr | 4.2e-5 |
| optimizer_momentum | $\beta_1, \beta_2 = 0.9, 0.95$ |
| lr schedule | cosine with warmup |
| batch_size | 64 |
| num_genomes | 50 |

Table 5: **Mistral training configuration.**

### A.4 FINETUNING EXPERIMENTS

We use the following datasets for finetuning:

- **NT Benchmark** (Dalla-Torre et al., 2023) consists of the following group of tasks histones, enhancers, promoters and splice sites.

- **Genomic Benchmarks** (Grešová et al., 2023) contains several datasets focused on regulatory element classification tasks across three organisms: human, mouse, and roundworm.

- **Genome Understanding Evaluation (GUE)** (Zhou et al., 2024) is a comprehensive multi-species benchmark containing 28 datasets across 7 genomic analysis tasks including promoter detection, transcription factor prediction, splice site detection, etc. with sequence lengths ranging from 70 to 1000 base pairs.

More details about the benchmarks can be found the corresponding original papers.

We finetune random and pretrained initializations of the chosen model using the configuration provided in Table 6. We found that final result was quite sensitive to learning rate, so we conducted a learning rate sweep over six different values and reported the highest result. Each task was run on 3 different folds, and the results were averaged. We used a validation holdout set for model selection and reported test scores for the epoch that corresponded to the highest score on validation set.

In our preliminary experiments, we found that max pooling performed better than cls / last pooling for randomly initialized models while maintaining performance for pretrained, so we used max pooling consistently across all experiments.

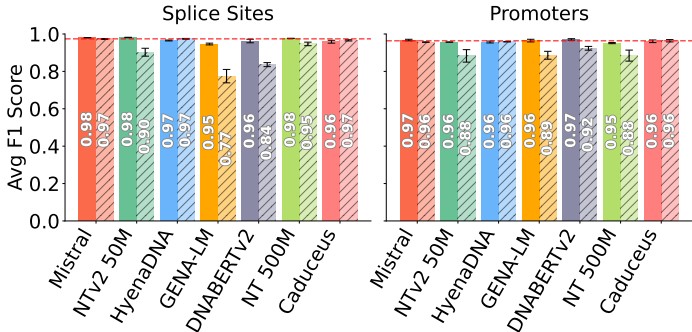

Figure 7: **NT Benchmark performance per subgroup for splice sites and promoter tasks.** Randomly initialized models are competitive with pretrained.

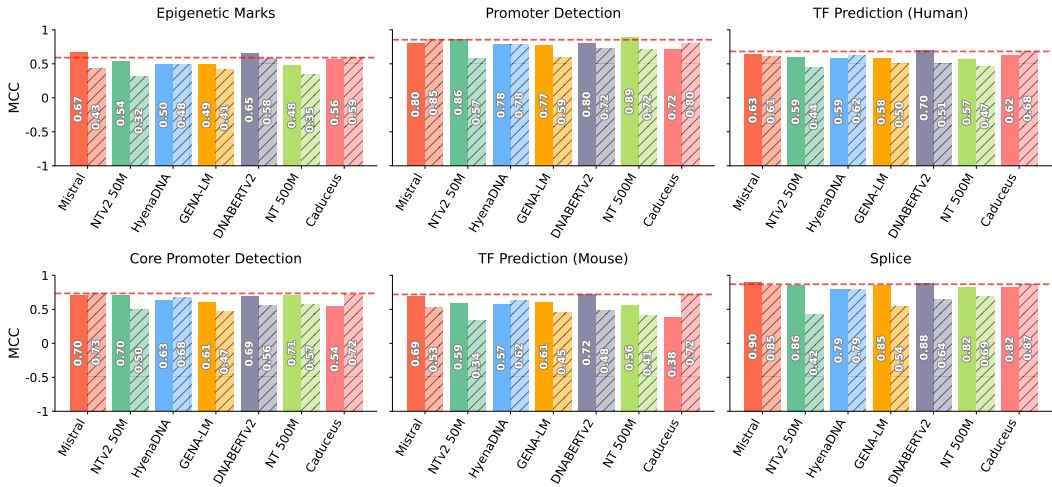

Figure 8: **GUE performance for each model.**

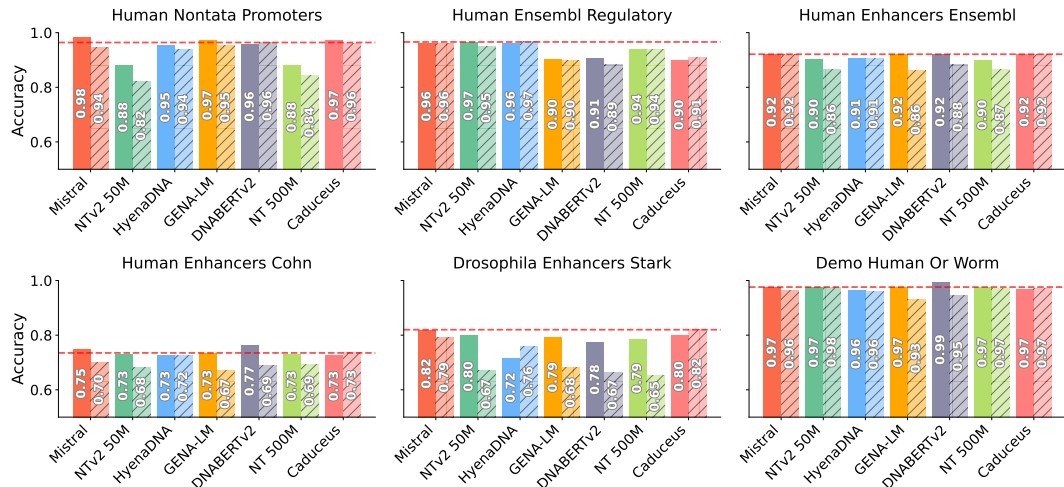

Figure 9: **Genomic Benchmarks performance for each model.**

| config | value |
|---|---|
| optimizer | AdamW |
| learning_rate | 1e-5, 3e-5, 5e-5, 8e-5, 1e-4, 3e-4 |
| weight_decay | 0 |
| optimizer_momentum | $\beta_1, \beta_2 = 0.9, 0.999$ |
| batch_size | 32 |
| lr schedule | cosine |
| epochs | 20 / 100 |

Table 6: **Hyperparameters for finetuning experiments.** For GUE we finetune for 20 epochs for NT Benchmark and Genomic Benchmarks we use 100 epochs.

## A.5 BIOTYPE CLASSIFICATION

Biotype task is a sequence classification task into nine different labels. Our dataset consists of a total of 19605 sequences. The detailed statistics for sequences belonging to each gene type is provided in Table 7. For the supervised training step, we perform a train-test split of 80% : 20% using stratification by class label. We use XGBoost with the hyperparameters provided in Table 9. All metrics are reported on the test set.

| Gene Type | Count | Avg Length | Max Length | Min Length |
|---|---|---|---|---|
| TEC | 1056 | 1613.26 | 18662 | 87 |
| lncRNA | 3000 | 32359.59 | 957949 | 87 |
| miRNA | 1879 | 81.89 | 180 | 41 |
| misc RNA | 2212 | 206.49 | 464 | 57 |
| processed pseudogene | 3000 | 798.02 | 12016 | 28 |
| protein coding | 3000 | 69971.51 | 2059620 | 159 |
| snRNA | 1901 | 110.46 | 328 | 50 |
| snoRNA | 943 | 118.86 | 791 | 55 |
| unprocessed pseudogene | 2614 | 5025.27 | 233909 | 28 |

Table 7: **Statistics of biotype genes.**

| Model | Embedding Dimensions |
|---|---|
| HyenaDNA | 64, 128, 256, 512, 1024, 2048, 4096 |
| Caduceus | 64, 128, 256, 512, 1024, 2048 |
| NT 50M | 64, 128, 256, 512, 1024, 2048, 4096 |
| DNABERTv2 | 48, 96, 192, 384, 768, 1152, 1536, 2304, 3072 |
| GENA-LM | 48, 96, 192, 384, 768, 1152, 1536, 2304, 3072 |
| NT 500M | 100, 320, 640, 1280, 2560, 3840 |
| Mistral | 64, 128, 256, 512, 1408, 2048, 4096 |

Table 8: **Embedding dimensions for biotype experiments.**

| config | value |
|---|---|
| objective | multi:softmax |
| num_classes | 9 |
| max_depth | 3 |
| learning_rate | 0.1 |
| n_estimators | 1000 |
| eval_metric | mlogloss |
| tree_method | hist |

Table 9: **Biotype XGBoost configuration.**

In addition, we also perform similar feature extraction experiments on the subset of GUE benchmark. Randomly initialized HyenaDNA with large embedding size outperforms all pretrained models.

| Task | HyenaDNA Random ED 2048 | Pretrained | | | | | |
|---|---|---|---|---|---|---|---|
| | | Mistral | HyenaDNA | NTv2 50M | GENA-LM | DNABERTv2 | NT 500M |
| H3 | **0.650** | 0.626 | 0.510 | 0.502 | 0.546 | 0.566 | 0.557 |
| H3K14ac | 0.275 | 0.227 | 0.190 | 0.272 | 0.208 | **0.338** | 0.220 |
| H3K36me3 | **0.408** | 0.267 | 0.252 | 0.330 | 0.321 | 0.397 | 0.308 |
| H3K4me1 | **0.320** | 0.224 | 0.211 | 0.275 | 0.244 | 0.295 | 0.267 |
| H3K4me2 | **0.265** | 0.243 | 0.186 | 0.176 | 0.218 | 0.185 | 0.245 |
| H3K4me3 | **0.207** | 0.126 | 0.105 | 0.147 | 0.113 | 0.189 | 0.121 |
| H3K79me3 | **0.522** | 0.428 | 0.367 | 0.463 | 0.437 | 0.520 | 0.406 |
| H3K9ac | **0.429** | 0.373 | 0.288 | 0.273 | 0.318 | 0.343 | 0.343 |
| H4 | **0.671** | 0.649 | 0.491 | 0.575 | 0.577 | 0.658 | 0.612 |
| H4AC | **0.282** | 0.227 | 0.202 | 0.227 | 0.200 | 0.259 | 0.225 |
| Average | **0.403** | 0.339 | 0.280 | 0.324 | 0.318 | 0.375 | 0.330 |

Table 10: **Feature Extraction on Histone Tasks from GUE.** Embeddings extracted from pretrained and randomly initialized models were used to train an XGBoost classifier. Randomly initialized HyenaDNA with $embed\_dim$ 2048 outperforms every pretrained model on every task except H314ac. MCC on test set is reported.

## A.6 ANCESTRY BENCHMARK

Each task is the sequence classification task with five labels, South Asian, European, African, American, East Asian. Each label is a superpopulation from 1000 Genomes dataset. We selected eleven different regions on chromosome with the length of 32K nucleotides, where each region corresponds to a different variant. The start indices with respect to the human reference genome used for sequence construction is provided in Table 11. Each task has 3202 samples.

Training involves two stages: embedding generation from the model of interest and supervised training on the embeddings with XGBoost. During the embedding generation step, sequence embeddings are constructed similarly to the biotype classification task. For the supervised training step, we split the dataset into train, validation and test set with sizes 72%, 8%, and 20% respectively. We use XGBoost with hyperparameters mentioned in Table 12. All metrics are reported on the test set and averaged over eleven tasks across each chromosome.

| chromosome | start position |
|------------|----------------|
| chr1 | 119478211 |
| chr3 | 2015011 |
| chr5 | 85769129 |
| chr7 | 74672986 |
| chr9 | 75197358 |
| chr11 | 62543311 |
| chr13 | 52182164 |
| chr15 | 45995594 |
| chr17 | 36628720 |
| chr19 | 24308808 |
| chr21 | 18354991 |

Table 11: **Sample chromosome positions.**

| config | value |
|--------|-------|
| objective | multi:softmax |
| num_class | 5 |
| max_depth | 3 |
| learning_rate | 0.1 |
| n_estimators | 1000 |
| colsample_bytree | 0.5 |
| eval_metric | mlogloss |
| tree_method | hist |
| early_stopping rounds | 100 |

Table 12: **Ancestry XGBoost configuration.**

### A.7 MUTATION SENSITITVITY EXPERIMENTS

In this section we provide the additional mutation sensitivity experiments for randomly initialized models.

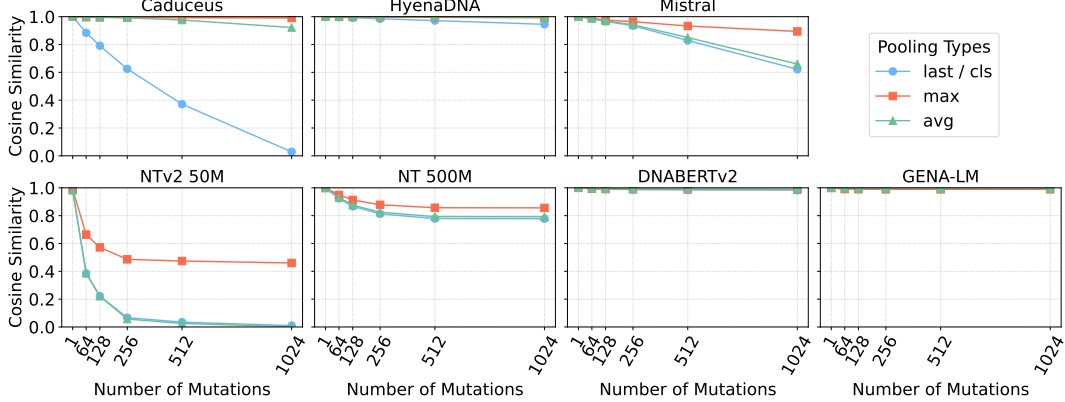

Figure 10: **Mutation sensitivity experiments with randomly initialized models.** Most of the models fail to capture the genomic variance.

### A.8 CLINVAR EXPERIMENTS

Each chunk used for ClinVar experiments consists of benign and pathogenic mutations. Three types of sequences are formed: reference sequence, sequence with benign mutations, and sequence with pathogenic mutations. The distribution of mutations in these chunks for all three genes is presented in Table 13.

| Chunk Index | TP53 | | BRCA2 | | CFTR | |
|-------------|------|---|-------|---|------|---|
| | Benign | Pathogenic | Benign | Pathogenic | Benign | Pathogenic |
| 1 | 122 | 27 | 138 | 46 | 32 | 18 |
| 2 | 60 | 61 | 268 | 74 | 19 | 11 |
| 3 | 51 | 50 | 187 | 57 | 32 | 30 |
| 4 | 76 | 42 | 35 | 18 | 9 | 13 |
| 5 | 38 | 10 | 37 | 6 | 7 | 11 |

Table 13: **Mutation Data Distribution by Gene and Chunk.** Distribution of benign and pathogenic mutations across different chunks for TP53, BRCA2, and CFTR genes.

## A.9 MODEL CHECKPOINTS

Checkpoints for all the pretrained models were obtained from Hugging Face. Table 14 provides detailed checkpoint IDs which can be loaded using the transformers library.

| Model | Checkpoint |
|---|---|
| NT 50M | InstaDeepAI/nucleotide-transformer-v2-50m-multi-species |
| NT 500M | InstaDeepAI/nucleotide-transformer-500m-1000g |
| Caduceus | kuleshov-group/caduceus-ps_seqlen-131k_d_model-256_n_layer-16 |
| HyenaDNA | LongSafari/hyenadna-tiny-1k-seqlen-hf |
| DNABERTv2 | zhihan1996/DNABERT-2-117M |
| GenaLM | AIRI-Institute/gena-lm-bert-base-t2t |

Table 14: **Checkpoints used for pretrained models.**

| Dataset | Metric | Pretrained Mistral | NT 50M | HyenaDNA | GENA-LM | DNABERTv2 | NT 500M | Caduceus | Random Mistral | NT 50M | HyenaDNA | GENA-LM | DNABERTv2 | NT 500M | Caduceus |
|---|---|---|---|---|---|---|---|---|---|---|---|---|---|---|---|
| H4ac | MCC | 0.702 ± 0.034 | 0.450 ± 0.001 | 0.476 ± 0.015 | 0.526 ± 0.006 | 0.652 ± 0.009 | 0.362 ± 0.003 | 0.610 ± 0.004 | 0.395 ± 0.032 | 0.254 ± 0.015 | 0.427 ± 0.008 | 0.382 ± 0.002 | 0.622 ± 0.006 | 0.245 ± 0.009 | 0.606 ± 0.007 |
| H4 | MCC | 0.802 ± 0.011 | 0.792 ± 0.000 | 0.783 ± 0.005 | 0.773 ± 0.013 | 0.809 ± 0.004 | 0.762 ± 0.001 | 0.778 ± 0.005 | 0.760 ± 0.012 | 0.593 ± 0.008 | 0.787 ± 0.003 | 0.619 ± 0.012 | 0.669 ± 0.006 | 0.629 ± 0.012 | 0.795 ± 0.007 |
| H3K9ac | MCC | 0.664 ± 0.004 | 0.540 ± 0.009 | 0.541 ± 0.004 | 0.551 ± 0.008 | 0.654 ± 0.004 | 0.473 ± 0.005 | 0.618 ± 0.005 | 0.470 ± 0.005 | 0.345 ± 0.007 | 0.532 ± 0.012 | 0.435 ± 0.018 | 0.593 ± 0.002 | 0.366 ± 0.016 | 0.615 ± 0.020 |
| H3K79me3 | MCC | 0.753 ± 0.005 | 0.596 ± 0.005 | 0.616 ± 0.005 | 0.633 ± 0.006 | 0.725 ± 0.006 | 0.574 ± 0.005 | 0.688 ± 0.020 | 0.666 ± 0.008 | 0.435 ± 0.010 | 0.576 ± 0.007 | 0.505 ± 0.001 | 0.670 ± 0.002 | 0.445 ± 0.015 | 0.685 ± 0.005 |
| H3K4me3 | MCC | 0.662 ± 0.016 | 0.340 ± 0.017 | 0.427 ± 0.004 | 0.487 ± 0.018 | 0.607 ± 0.002 | 0.259 ± 0.016 | 0.565 ± 0.008 | 0.444 ± 0.018 | 0.168 ± 0.004 | 0.354 ± 0.009 | 0.223 ± 0.129 | 0.606 ± 0.014 | 0.162 ± 0.003 | 0.555 ± 0.002 |
| H3K4me2 | MCC | 0.574 ± 0.024 | 0.303 ± 0.004 | 0.367 ± 0.012 | 0.445 ± 0.018 | 0.564 ± 0.005 | 0.289 ± 0.012 | 0.463 ± 0.007 | 0.270 ± 0.001 | 0.207 ± 0.009 | 0.302 ± 0.006 | 0.316 ± 0.010 | 0.511 ± 0.000 | 0.220 ± 0.004 | 0.493 ± 0.002 |
| H3K4me1 | MCC | 0.603 ± 0.007 | 0.518 ± 0.002 | 0.441 ± 0.003 | 0.468 ± 0.005 | 0.585 ± 0.002 | 0.397 ± 0.013 | 0.493 ± 0.015 | 0.442 ± 0.027 | 0.260 ± 0.009 | 0.421 ± 0.004 | 0.334 ± 0.006 | 0.499 ± 0.007 | 0.254 ± 0.008 | 0.534 ± 0.003 |
| H3K36me3 | MCC | 0.725 ± 0.007 | 0.581 ± 0.003 | 0.538 ± 0.008 | 0.555 ± 0.011 | 0.676 ± 0.011 | 0.469 ± 0.008 | 0.613 ± 0.022 | 0.573 ± 0.014 | 0.371 ± 0.018 | 0.495 ± 0.004 | 0.416 ± 0.008 | 0.611 ± 0.016 | 0.335 ± 0.011 | 0.625 ± 0.013 |
| H3K14ac | MCC | 0.724 ± 0.011 | 0.516 ± 0.013 | 0.516 ± 0.007 | 0.587 ± 0.009 | 0.659 ± 0.006 | 0.396 ± 0.024 | 0.615 ± 0.008 | 0.578 ± 0.013 | 0.281 ± 0.009 | 0.462 ± 0.007 | 0.403 ± 0.016 | 0.641 ± 0.003 | 0.259 ± 0.005 | 0.637 ± 0.011 |
| H3 | MCC | 0.809 ± 0.003 | 0.769 ± 0.006 | 0.778 ± 0.007 | 0.776 ± 0.005 | 0.826 ± 0.012 | 0.741 ± 0.013 | 0.802 ± 0.012 | 0.679 ± 0.012 | 0.573 ± 0.008 | 0.770 ± 0.005 | 0.656 ± 0.001 | 0.742 ± 0.009 | 0.597 ± 0.008 | 0.795 ± 0.002 |
| enhancers_types | MCC | 0.528 ± 0.006 | 0.456 ± 0.008 | 0.424 ± 0.028 | 0.477 ± 0.012 | 0.469 ± 0.002 | 0.433 ± 0.023 | 0.487 ± 0.018 | 0.441 ± 0.028 | 0.438 ± 0.025 | 0.412 ± 0.029 | 0.474 ± 0.013 | 0.488 ± 0.019 | 0.454 ± 0.008 | 0.451 ± 0.011 |
| enhancers | MCC | 0.557 ± 0.003 | 0.576 ± 0.010 | 0.552 ± 0.009 | 0.569 ± 0.010 | 0.536 ± 0.009 | 0.557 ± 0.027 | 0.527 ± 0.012 | 0.561 ± 0.004 | 0.523 ± 0.016 | 0.556 ± 0.022 | 0.545 ± 0.015 | 0.529 ± 0.002 | 0.540 ± 0.011 | 0.523 ± 0.024 |
| splice_sites_all | F1 Score | 0.980 ± 0.000 | 0.980 ± 0.000 | 0.962 ± 0.006 | 0.941 ± 0.002 | 0.950 ± 0.001 | 0.977 ± 0.002 | 0.959 ± 0.003 | 0.977 ± 0.000 | 0.908 ± 0.015 | 0.975 ± 0.001 | 0.726 ± 0.003 | 0.835 ± 0.005 | 0.951 ± 0.003 | 0.967 ± 0.003 |
| splice_sites_donors | F1 Score | 0.979 ± 0.001 | 0.981 ± 0.002 | 0.966 ± 0.003 | 0.945 ± 0.003 | 0.964 ± 0.002 | 0.977 ± 0.001 | 0.949 ± 0.009 | 0.971 ± 0.001 | 0.925 ± 0.006 | 0.971 ± 0.001 | 0.811 ± 0.008 | 0.823 ± 0.003 | 0.956 ± 0.003 | 0.962 ± 0.004 |
| splice_sites_acceptors | F1 Score | 0.981 ± 0.001 | 0.983 ± 0.001 | 0.968 ± 0.002 | 0.953 ± 0.002 | 0.973 ± 0.002 | 0.975 ± 0.002 | 0.968 ± 0.003 | 0.972 ± 0.002 | 0.874 ± 0.017 | 0.976 ± 0.003 | 0.787 ± 0.000 | 0.850 ± 0.014 | 0.934 ± 0.002 | 0.970 ± 0.001 |
| promoter_tata | F1 Score | 0.962 ± 0.007 | 0.956 ± 0.003 | 0.961 ± 0.000 | 0.955 ± 0.003 | 0.965 ± 0.004 | 0.947 ± 0.000 | 0.951 ± 0.007 | 0.958 ± 0.001 | 0.836 ± 0.009 | 0.960 ± 0.004 | 0.856 ± 0.012 | 0.909 ± 0.003 | 0.843 ± 0.002 | 0.955 ± 0.007 |
| promoter_no_tata | F1 Score | 0.970 ± 0.003 | 0.958 ± 0.002 | 0.960 ± 0.001 | 0.970 ± 0.002 | 0.975 ± 0.000 | 0.954 ± 0.001 | 0.967 ± 0.002 | 0.957 ± 0.001 | 0.906 ± 0.002 | 0.958 ± 0.001 | 0.899 ± 0.003 | 0.929 ± 0.001 | 0.903 ± 0.002 | 0.969 ± 0.001 |
| promoter_all | F1 Score | 0.971 ± 0.002 | 0.957 ± 0.002 | 0.960 ± 0.001 | 0.969 ± 0.001 | 0.973 ± 0.001 | 0.953 ± 0.000 | 0.965 ± 0.000 | 0.955 ± 0.001 | 0.907 ± 0.003 | 0.958 ± 0.001 | 0.903 ± 0.003 | 0.932 ± 0.001 | 0.908 ± 0.003 | 0.968 ± 0.000 |

Table 15: **Pretrained and randomly initialized models performance on NT Benchmark.**

