# OpenReview forum: "Genomic Foundationless Models: Pretraining Does Not Promise Performance"
_ICLR.cc/2025/Conference — ICLR 2025 Conference Withdrawn Submission_

### Official Review · Reviewer_KtQq · 2024-11-02

**Soundness:** 2
**Presentation:** 2
**Contribution:** 2
**Rating:** 3
**Confidence:** 4

**Summary:**

This paper investigates the performance of genomic foundationless models, particularly focusing on the implications of pretraining. While the topic is pertinent, the paper presents several critical issues that undermine its contributions and clarity.

**Strengths:**

- The exploration of pretraining in genomic models is an important area of research, especially given the growing interest in applying deep learning to genomics.
- Potential Impact: Understanding the limitations of pretraining in this context can provide valuable insights for future research.

**Weaknesses:**

- The findings largely reiterate known limitations of pretraining without offering new insights or approaches. The paper does not present a compelling argument for how its contributions advance the field.

- The experimental validation lacks rigor. There is a failure to adequately compare the proposed models with established methods, making it difficult to evaluate the significance of the results. A large number of genetic tasks that have been established have not been mentioned and tested for relevance [1].

- The conclusions drawn from the experiments are overly broad and not sufficiently supported by the data presented. The authors claim that pretraining does not promise performance without adequately accounting for various contextual factors.

[1] GenBench: A Benchmarking Suite for Systematic Evaluation of Genomic Foundation Models

**Questions:**

- It's not that pre-training doesn't work. It's that gene sequences haven't found a more appropriate pre-training strategy.

- It doesn't make sense to me to disregard the EVO model, which has the largest pre-training scale, doesn't it?

---

> ### Author Response · Authors · 2024-11-27
>
> First of all, we have comments about some of your specific quotes:
>
> >This paper investigates the performance of genomic **foundationless** models, particularly focusing on the implications of pretraining.
>
> We investigate the performance of genomic foundation models, not foundationless. We call the them foundationless as a result of our analysis, where we discover that they lack any significant genomic understanding.
>
> >There is a failure to adequately compare the **proposed models** with established methods, making it difficult to evaluate the significance of the results.
>
> Our paper does not propose any models but rather studies the existing ones.

---

> ### Author Response · Authors · 2024-11-27
>
> > **W1.** The findings largely reiterate known limitations of pretraining without offering new insights or approaches. The paper does not present a compelling argument for how its contributions advance the field.
>
> We respectfully disagree that our findings merely reiterate known limitations. If you claim that our findings reiterate largely known limitations you should provide references for other papers that discuss those, which you do not.
>
> While [Tang et al., 2024] recently examined some genomic language model limitations, our study differs substantially in scope and methodology:
>
> 1. We evaluate 7 different genomic foundation models (GFMs), including several released in 2024, while [Tang et al., 2024] analyzed 4 older models.
>
> 2. We systematically compare pretrained models against their randomly initialized counterparts.
>
> 3. We are first to build a baseline with random model's embeddings as features and show that these embeddings are competitive with pretrained ones.
>
> 4. Our variant-based analyses (Section 3.3) reveal previously undocumented limitations in GFMs' ability to detect clinically relevant mutations.
>
> 5. We provide extensive ablation studies on embedding dimensions and tokenization strategies that give actionable insights for model improvements.
>
> 6. Beyond identifying limitations, we demonstrate that simple architectural modifications like changing tokenizers and increasing embedding dimensions can allow randomly initialized models to outperform pretrained ones (Section 3.2).
>
> Moreover, even the work of [Tang et al., 2024] was released only a few months ago, which directly contradicts the notion that these results are "well-known" limitations.
>
> [Tang et al., 2024] Evaluating the representational power of pre-trained DNA language models for regulatory genomics

---

> ### Author Response · Authors · 2024-11-27
>
> > **W2.** The experimental validation lacks rigor. There is a failure to adequately compare the proposed models with established methods, making it difficult to evaluate the significance of the results.
>
> We respectfully disagree with this statement. This claim is also unsubstantiated as you didn't provide any specific examples in which our methodology is unclear.
>
> In contrast, we have made significant efforts to ensure robust validation and reproducibility. Our experimental validation includes:
>
> * Comprehensive evaluation of models over three different groups of tasks: finetuning, feature extraction, and variant sensitivity analysis. We believe this is the biggest evaluation of Genomic Foundation Models up to the date.
>
> * Statistical analysis with error bars over 3 folds for NT Benchmark (nearly 5,000 experiments in total) and 11 folds for ancestry experiments.
>
> * Variant sensitivity results averaged over 5 sample strings across 5 chromosomes (25 different sequences in total) to ensure reliable conclusions.
>
> * Detailed ablation studies examining the effects of embedding dimensions and tokenizer choice in the biotype classification task.
>
> * Complete hyperparameter configurations for all experiments documented in the Appendix.
>
> * Systematic learning rate sweeps over 6 different values for each task to ensure optimal performance.
>
> * We also provide the code for the experiments through an anonymous link.
>
> * In addition, we also provide more information that will help in reproducibility of our work in the Reproducibility Statement section.

---

> ### Author Response · Authors · 2024-11-27
>
> > A large number of genetic tasks that have been established have not been mentioned and tested for relevance [1].
>
> In the last revision, we added more than 30 tasks from GUE and Genomics Benchmarks, thus making our analysis significantly more comprehensive. We believe this is the biggest evaluation of Genomic Foundation Models up to the date. Our evaluation now includes:
>
> 1. Finetuning Evaluation (52 tasks) across three benchmark suites:
>   * NT Benchmark
>   * GUE Benchmark
>   * Genomics Benchmark
>
> 2. Feature Extraction Tasks:
>   * Biotype classification (9 gene types)
>   * 10 distinct tasks from GUE
>
> 3. Variant Analysis:
>   * Mutation sensitivity experiments
>   * ClinVar-based evaluation of pathogenic variants
>   * Ancestry prediction across 11 genomic regions
>
> In total, our paper now evaluates over 52 distinct classification tasks, 11 feature extraction tasks, and 3 benchmarks for genomic variation.
>
> Moreover, GenBench is just an ensemble of currently existing tasks from other benchmarks and it is unclear how these will invalidate our results.

---

> ### Author Response · Authors · 2024-11-27
>
> > **W3.** The conclusions drawn from the experiments are overly broad and not sufficiently supported by the data presented. The authors claim that pretraining does not promise performance without adequately accounting for various contextual factors.
>
> Our conclusions are carefully drawn from extensive empirical evidence across many benchmarks. It is also not exactly clear what various contextual factors you are referring to.
>
> In our comprehensive analysis we consider various factors such as:
>
> 1. Model Diversity: We evaluate 7 different architectures including:
> * Decoder-only models (Mistral, HyenaDNA, Caduceus)
> * Encoder-only models (NT, GENA-LM, DNABERTv2)
> * Different scales (from 450K to 500M parameters)
> * Various tokenization strategies (character, k-mer, BPE)
>
> 2. Training Approaches:
> * Different pretraining objectives
> * Multiple pretraining datasets (HRG, 1000G, multispecies)
> * Various sequence lengths (from 128 to 131K)
>
> 3. Evaluation Methods:
> * Finetuning performance: NT Benchmark. GUE, Genomic Benchmarks
> * Feature extraction capabilities biotype classification and GUE
> * Genomic variance: mutation sensitivity, ancestry prediction, ClinVar analysis
>
> Despite such big differences in the analyzed models we yield very consistent conclusions that current pretraining in genomics does not offer substantial improvement.

---

> ### Author Response · Authors · 2024-11-27
>
> > **Q1.** It's not that pre-training doesn't work. It's that gene sequences haven't found a more appropriate pre-training strategy.
>
> In our paper, including the abstract, we are explicitly saying that there is a need to critically rethink the pretraining approaches in genomics. In the end of the introduction section we also provide some of the potential ways to improve the effectiveness of the pretraining (improving the tokenizers, etc).

---

> ### Author Response · Authors · 2024-11-27
>
> > **Q2.** It doesn't make sense to me to disregard the EVO model, which has the largest pre-training scale, doesn't it?
>
> In our paper we explicitly say as to why we did not include the EVO model:
>
> "We excluded the EVO model from our analysis as it was trained on bacterial genomes and performed poorly in our preliminary tests on the Nucleotide Transformer Benchmark."

---

### Official Review · Reviewer_Q34j · 2024-11-02

**Soundness:** 3
**Presentation:** 4
**Contribution:** 3
**Rating:** 6
**Confidence:** 4

**Summary:**

The paper critically evaluates the assumption that pretraining offers significant advantages in Genomic Foundation Models (GFMs). Through extensive comparisons, the authors show that randomly initialized models can achieve performance comparable to or exceeding that of pretrained models across a variety of genomics tasks.

**Strengths:**

1. The study includes a detailed analysis across several GFMs with random initialization methods, providing a broad view of the pretrained effect on genomic benchmarks.
2. The observation that pretrained models often fail to capture clinically relevant mutations, particularly for tasks that require sensitivity to sequence variation.
3. The authors conducted extensive and convincing experiments on genomic datasets to validate their hypotheses.

**Weaknesses:**

1. Lack of further analysis on model behavior. For instance, why do random models outperform pretrained models on some tasks? Why does Mistral perform better in mutation sensitivity analysis?
2. The experiment on ClinVar uses a limited sample size (only one gene), which may limit the generalizability of the conclusion that pretrained models lack sensitivity to mutations.
3. The single nucleotide sensitivity analysis lacks comparative experiments. There is no direct comparison between random and pretrained models in the mutation sensitivity analysis and the ClinVar experiments.

**Questions:**

See Weaknesses.

---

> ### Author Response · Authors · 2024-11-27
>
> > **W1.** Lack of further analysis on model behavior. For instance, why do random models outperform pretrained models on some tasks?
>
> We think that main reason is that current pretraining approaches are fundamentally unsuitable for genomic data and so far they simply copy NLP methods without proper adaptation to genomics domain. For example:
>
> 1. NT 500M pretrained on variant data uses masked language modeling with 15% masking probability, yet real genomic mutations occur at only ~0.5% probability.
>
> 2. k-mer and BPE tokenizers generally work worse than character tokenizers as have been shown in our biotype experiments. These tokenizers are also not suitable for genomic variation tasks as they aggregate multiple tokens into a single nucleotide.
>
> 3. Misalignment between pretraining objective and downstream tasks. Even though the perplexity decreases during pretraining there is little to no correlation of perplexity to downstream tasks.

---

> ### Author Response · Authors · 2024-11-27
>
> > Why does Mistral perform better in mutation sensitivity analysis?
>
> The model leverages several advanced architectural design choices, which are not always present in other GFMs:
>
> * RoPE positional embeddings that have shown strong performance in language modeling
>
> * Dense attention mechanism which, based on our results, still outperforms state-space models like HyenaDNA for genomic tasks
>
> * Character-level tokenization, which aligns with recent findings showing character tokenizers are more effective for genomic data [Lindsey et al., 2024]
>
> * Largest embedding dimension compared to other models
>
> Moreover, it has also been trained on 1000G data, where we sampled the variants across 50 samples. On average during training among 4k tokens 20 (0.5%) tokens are mutated. It aligns better with character tokenizer and not with kmer or BPE, and this is the only model that was pretrained on nucleotide + variant data.
>
> We updated the paper to make sure these details are reflected in the new revision.

---

> ### Author Response · Authors · 2024-11-27
>
> > **W2.** The experiment on ClinVar uses a limited sample size (only one gene), which may limit the generalizability of the conclusion that pretrained models lack sensitivity to mutations.
>
> We provided additional experiments on two more genes: BRCA2 and CFTR. As you can see from results, the conclusion that GFMs are insensitive to mutations generalize on the newly added genes as well.
>
> We updated the manuscript to reflect these new experiments.

---

> ### Author Response · Authors · 2024-11-27
>
> > **W3.** The single nucleotide sensitivity analysis lacks comparative experiments. There is no direct comparison between random and pretrained models in the mutation sensitivity analysis and the ClinVar experiments.
>
> We added the results for random models for mutation sensitivity into the manuscript, Figure 9 in the Appendix. Random embeddings from Caduceus and NTv2 50M are more sensitive to changes in a sequence, random embeddings for Mistral are less sensitive. This is consistent with results from other experiments where we consistently see random Caduceus outperforming pretrained counterparts, and pretrained Mistral outperforming random Mistral.

---

### Official Review · Reviewer_h8wc · 2024-11-03

**Soundness:** 2
**Presentation:** 3
**Contribution:** 2
**Rating:** 3
**Confidence:** 5

**Summary:**

This work benchmarks the effectiveness of the Genomics Foundation Models (GFM) compared to randomly initialized models. Their experiments indicate that GFMs offer limited or even no advantages over randomly initialized models on many tasks.

**Strengths:**

1. The paper is well-written and easy to follow.
2. The experiments are comprehensive and well-presented, except for a few ones (see Weaknesses).
3. The mutation experiments are very interesting.

**Weaknesses:**

1. Findings 1 is not well-supported
    - Figure 1 is missing leading. Given that different GFMs differ significantly in the number of parameters (e.g., 450k v.s. 500M), I don't understand what the authors aim to convey with this figure. A randomly initialized large model achieving better performance than a pre-trained small model does not suggest the pre-training is not helpful.
    - A pairwise comparison of the pre-trained and random versions of the same model, as shown in Figure 2, is much more meaningful. In fact, as shown in Figure 2, the pre-trained model performs better than the random counterparts in most cases, which actually suggests the effectiveness of pre-training. The improvement of pre-training over randomized is more significant on more distinguishable datasets like histone (i.e., different models achieve distinct performances) and less significant on promoter/enhancer. I would attribute this insignificance mainly to the datasets (See below).

2. As shown in Figure 2, all the models ranging from 450k to 500M parameters achieve near-identical performance on many datasets of the NT benchmark, especially on promoter and enhancer. This indicates this benchmark may not be suitable for differentiating different models' capabilities. If the dataset could not differentiate different pre-trained models, it's hard to believe it can effectively differentiate pre-trained/random models. I would suggest more experiments on other benchmarks/datasets that can clearly differentiate the performance of different models. You may consider the datasets used in HyeneDNA (species classification), DNABERT-2 (GUE), and Caduceus (Genomics Benchmark).


Overall, I appreciate the authors' efforts in conducting this amount of experiments. Yet the results are not convincing, and the main conclusion (findings 1) is not well-supported. I will consider increasing my scores if more convincing results are presented.

**Questions:**

1. For NT, why use the 1000G instead of NTv2-multispecies? The v2 one is much better.
2. It's nice to compare the models in both finetuning and feature extraction. Why don't you use the same datasets for finetune/feature extraction? I would suggest the same feature extraction experiments on the same selected portion of the NT benchmark, like the histone datasets, to provide more convincing observations on the feature extraction of GFMs.

---

> ### Author Response · Authors · 2024-11-27
>
> Thank you very much for your valuable feedback below you can find a summary of edits based on your suggestions:
>
> * We added experiments on GUE and Genomics Benchmark showing that our claims regarding the competitiveness of the randomly initialized models still hold on new benchmarks.
>
> * We revised Finding 1 and added Finding 2 together with discussion about the comparisons of pretraining gains in genomics vs computer vision and NLP. We clarified that the gains in genomics are quite marginal compared to computer vision and NLP.
>
> * We clarified that we do not claim that all randomly initialized models are always competitive, but rather we are able to identify a few randomly initialized models that surpass many pretrained ones.
>
> * We clarified in Finding 1 that the competitiveness of randomly initialized models is not a function of the model size. In fact the best randomly initialized model is Caduceus with only 8M parameters.

---

> ### Author Response · Authors · 2024-11-27
>
> > **W1a.** Figure 1 is missing leading. Given that different GFMs differ significantly in the number of parameters (e.g., 450k v.s. 500M), I don't understand what the authors aim to convey with this figure. **A randomly initialized large model achieving better performance than a pre-trained small model does not suggest the pre-training is not helpful.**
>
> Thank you for pointing that out, we edited the old Figure 1 (it is now Figure 2 in the new revision). Now we display the differences in performance between the pretrained models and the best random model to better highlight the competitiveness of the random models.
>
> Moreover, for each of the datasets we also include the name of the best random model. As you can see in many cases the best random model is Caduceus that has just 8M parameters: this is 60 smaller than NT 500M model, while random Caduceus outperforms pretrained NT 500M by a large margin on many datasets.
>
> We also updated the Finding 1 accordingly to clarify that competitiveness is not a function of the model size.

---

> ### Author Response · Authors · 2024-11-27
>
> > **W1b.** A pairwise comparison of the pre-trained and random versions of the same model, as shown in Figure 2, is much more meaningful. In fact, as shown in Figure 2, the pre-trained model performs better than the random counterparts in most cases, which actually suggests the effectiveness of pre-training. The improvement of pre-training over randomized is more significant on more distinguishable datasets like histone (i.e., different models achieve distinct performances) and less significant on promoter/enhancer. I would attribute this insignificance mainly to the datasets (See below).
>
> Thank you for pointing that out, as per your request we include additional benchmarks such as GUE and Genomic Benchmarks. Interestingly these new datasets, and particularly GUE, only reinforce our initial conclusions about competitiveness of pretrained models. The results on these new benchmarks, especially GUE, only solidify our initial findings from NT Benchmark. We also include additional pairwise comparisons plots for GUE and Genomic Benchmarks in the Appendix.
>
> Moreover, we added the discussion about the comparison of pretraining gains in genomics vs computer vision and NLP. We highlight that in NLP and CV pretraining gives double digit gains, while in genomics small randomly initialized models often outperform the big pretrained models and in the best case pretrained models have a very marginal advantage over the randomly models, typically 2-3%. These marginal gains do not justify high computational costs required for conducting pretraining.

---

> ### Author Response · Authors · 2024-11-27
>
> > **W2.** As shown in Figure 2, all the models ranging from 450k to 500M parameters achieve near-identical performance on many datasets of the NT benchmark, especially on promoter and enhancer. This indicates this benchmark may not be suitable for differentiating different models' capabilities. If the dataset could not differentiate different pre-trained models, it's hard to believe it can effectively differentiate pre-trained/random models. I would suggest more experiments on other benchmarks/datasets that can clearly differentiate the performance of different models. You may consider the datasets used in HyeneDNA (species classification), DNABERT-2 (GUE), and Caduceus (Genomics Benchmark).
>
> Thanks for your suggestion. We agree with you that splice sites and promoter tasks on NT Benchmark do not allow to differentiate between models, so we moved them into Appendix and instead display the results for GUE and Genomic Benchmarks in the main part of the paper. However, on GUE we found that small randomly initialized Caduceus beats large pretrained models. Similar pattern observed for Genomic Benchmarks. Therefore, our original conclusion still holds.

---

> ### Author Response · Authors · 2024-11-27
>
> > **Q1:** For NT, why use the 1000G instead of NTv2-multispecies? The v2 one is much better.
>
> Actually, we used NTv2 50M Multispecies in all our experiments. We updated the model name everywhere from NT 50M → NTv2 50M.
>
> We also wanted to achieve both diversity in parameter count and have at least one existing model trained on human variance data. Therefore, we selected NT 500M 1000G and not NTv2 500M which was trained on Multispecies. Moreover, we have not found a significant advantage of NTv2 500M over NTv2 50M, so we decided to use a smaller model.

---

> ### Author Response · Authors · 2024-11-27
>
> > **Q2:** It's nice to compare the models in both finetuning and feature extraction. Why don't you use the same datasets for finetune/feature extraction? I would suggest the same feature extraction experiments on the same selected portion of the NT benchmark, like the histone datasets, to provide more convincing observations on the feature extraction of GFMs.
>
> We additionally ran feature extraction experiments on 10 histone modification tasks from GUE benchmark. The new feature extraction results for GUE tasks are presented in the Table 10 in the Appendix. There was a significant variance in pretrained GFM performance, however randomly initialized HyenaDNA with embedding dimension of 2048 was best for 9/10 tasks and also on average (0.403 MCC vs second-best pretrained DNABERTv2 with 0.375).
>
> It is worth noting that finetuned models are significantly better in general, which indicates that the current generation of GFM is not able to produce meaningful representations and therefore cannot be used as feature extractors without finetuning on downstream tasks.

---

### Official Review · Reviewer_M5MV · 2024-11-03

**Soundness:** 2
**Presentation:** 3
**Contribution:** 2
**Rating:** 5
**Confidence:** 3

**Summary:**

The paper questions the value of pretraining in genomic foundation models (GFMs), demonstrating that randomly initialized models can often match or even surpass pretrained GFMs on several tasks.

**Strengths:**

1. The paper proposes an interesting perspective in challenging the assumed superiority of pretrained models in genomics;
2. The experiments were thorough in comparing pretrained models with randomly initialized counterparts across various genomic tasks;
3. The writing is clear and easy to follow.

**Weaknesses:**

1. Although the results suggest current methods may not work, the paper could explore alternative pretraining objectives or paradigms that might better align with biological complexity.
2. There lacks an explanation or analysis on why specific models, such as Mistral or DNABERTv2, might perform better or worse on certain tasks, which would provide better insight.

**Questions:**

1. Could different pretraining tasks or loss functions enhance mutation sensitivity or biotype classification, specifically for detecting single-nucleotide variants?
2. The paper indicates certain tokenization changes improved performance. Would further model-specific optimizations, such as vocabulary adaptations or alternate pooling strategies, yield better results?
3. Given that pretrained models in other domains often benefit from more extensive data and compute, is there a threshold or scaling that could make pretraining beneficial for GFMs?

---

> ### Author Response · Authors · 2024-11-27
>
> We thank the reviewer for the provided feedback and we are happy to address your concerns.
>
> > **W1.** Although the results suggest current methods may not work, the paper could explore alternative pretraining objectives or paradigms that might better align with biological complexity.
>
> While we agree this is an important direction, we believe it falls outside the scope of our current work, which focuses on providing a comprehensive evaluation of existing pretraining approaches in genomics. Our paper presents an extensive analysis covering:
> 1. Seven different genomic foundation models with diverse architectures, training objectives, and pretraining data.
> 2. Multiple evaluation paradigms:
> Finetuning experiments on the NT Benchmark, GUE benchmark, Genomic Benchmarks, totaling ~52 finetuning tasks, and in the ballpark of around 10,000 experiment runs.
> Feature extraction through biotype classification and GUE tasks.
> Genomic variation sensitivity through ancestry prediction, mutation analysis and ClinVAR.
> We also think that exploring the alternative approaches in the same work would dilute the paper’s core message that pretraining does not provide substantial benefits in genomics. This paper's focus is on running a large number of experiments to provide enough evidence for it.
>
> Moreover, we are already working on the follow up work that will explore the different tokenization strategies more tailored for genomics to make the pretraining more effective.

---

> ### Author Response · Authors · 2024-11-27
>
> > **W2.** There lacks an explanation or analysis on why specific models, such as Mistral or DNABERTv2, might perform better or worse on certain tasks, which would provide better insight.
>
> We appreciate the reviewer's suggestion. We attribute the success of Mistral to the following reasons:
>
> 1. Mistral has the most recent and advanced architecture recipe which has:
>    * RoPE positional embeddings that have shown strong performance in language modeling. In contrast many BERT-style models such as DNABERTv2 use ALIBI.
>    * Dense attention mechanism, which generally outperforms feature extraction mechanisms used in state-space models.
>    * Character-level tokenization, which aligns with recent findings showing character tokenizers are more effective for genomic data [Lindsey et al., 2024]
>    * The largest embedding dimension compared to other models, which as shown in the biotype section is really helpful.
>
> Many of these components are missing from other models and, hence, advantage of Mistral.
>
> 2. DNABERTv2's strong performance can be attributed to:
>    * Training on multispecies data, which appears to provide better generalization.
>    * BPE tokenization strategy that outperforms k-mer based approaches used in models like NTv2 50M and NT 500M.
>
> In addition to this analysis, we have already provided the following explanations in the manuscript:
>
> * We demonstrate that character-level tokenization significantly improves performance of random encoder-only models on biotype. We explain that this happens because character tokenization reduces the search space to just 4 tokens (A, C, G, T). Mistral uses character-level tokenization.
>
> * We conjecture that NT 500M's poor performance on mutation cases, despite being trained on 1000G data, is likely due to its masked language modeling objective with 15% masking probability being much higher than the 0.5% mutation probability in real sequences.
>
> We have revised our paper accordingly to include these explanations.
>
> [Lindsey et al., 2024] A Comparison of Tokenization Impact in Attention Based and State Space Genomic Language Models

---

> ### Author Response · Authors · 2024-11-27
>
> > **Q1.** Could different pretraining tasks or loss functions enhance mutation sensitivity or biotype classification, specifically for detecting single-nucleotide variants?
>
> For mutation sensitivity we identified several fundamental challenges:
>
> 1. Sparsity Problem: mutations are extremely sparse in genomic sequences (0.5% of positions). In our preliminary experiments on pretraining, simply adjusting token weights in the cross-entropy loss for next token prediction to emphasize mutation tokens did not improve downstream sensitivity.
>
> 2. Scaling Limitations: Current approaches cannot effectively scale beyond ~3000 genomes (which is how many in 1000G dataset) due to the computational constraints. Dataset consisting of 3000 human genomes corresponds to roughly **9 Trillion** tokens, considering the single nucleotide resolution tokenizer, and scaling up the data would also require scaling the model size in parallel which does not make it feasible due to compute limits.
>
> In case of decoder and encoder models the most likely way to improve the mutation sensitivity would be to work on the improved tokenizer that will incorporate special mutation tokens that would indicate the start or end of the functional region or some other similar ideas.
>
> However, we believe a more promising direction is to move away from full sequence pretraining on continuous chunks of nucleotides to a 'bag of variants' approach that processes only variants and their positions. This solution would be much more computationally efficient, better aligned with the sparse nature of mutations, and more scalable to large numbers of samples.
>
> Regarding the biotype, Table 7 in the Appendix presents data statistics indicating that there are many long sequences in this task, specifically those longer than 5000 nucleotides. This implies that the model must effectively process and embed long sequences. Although HyenaDNA and Caduceus can handle these sequence lengths, they struggle to extract useful representations as the signal gets lost. Further work is necessary to improve the extraction of useful representations from long sequences, which potentially will improve the model performance on biotype classification.

---

> ### Author Response · Authors · 2024-11-27
>
> > **Q2.** The paper indicates certain tokenization changes improved performance. Would further model-specific optimizations, such as vocabulary adaptations or alternate pooling strategies, yield better results?
>
> Yes, we think a lot of improvement can come from improving tokenization by introducing special tokens that would indicate start / end of coding region or gene and similar. Furthermore, we think one potential strategy would be to use different tokenizers for coding vs non-coding regions and similar ideas. For example, you can use character tokenizer in the areas which you deem more important and use BPE in less important areas that would greatly reduce the number of tokens corresponding to less important areas and thus will make the signal less sparse.
>
> Regarding the pooling we've mainly explored three different types: max, avg, and cls / last. We have found that max pooling was mostly giving the best performance, but we also think better methods can be developed.

---

> ### Author Response · Authors · 2024-11-27
>
> > **Q3.** Given that pretrained models in other domains often benefit from more extensive data and compute, is there a threshold or scaling that could make pretraining beneficial for GFMs?
>
> While scaling has been beneficial for pretrained models in other domains, our analysis suggests that simply scaling current GFM approaches may not yield similar benefits in genomics for several reasons:
>
> 1. While pretraining improves perplexity, these improvements in perplexity don't currently translate to better performance on downstream tasks.
>
> 2. Current genomic datasets, despite their huge size, are highly redundant and may not provide the enough signal density needed for effective pretraining. Better data curation could help to greatly improve the quality of the training data.
>
> 3. More improvement needs to be done on tokenization and pretraining objectives to help models learn more efficiently. This will also partially address #1.

---

> > ### Comment · Reviewer_M5MV · 2024-12-03
> > **Response to Rebuttal**
> >
> > We thank the authors' effort in their rebuttal. While they have solved some of my questions, there are remaining concerns, e.g., the paper lacks deeper insights into the difference between models. Issues with scalability and biological relevance are still partly unresolved. So I decide to maintain my score.

---

> > > ### Author Response · Authors · 2024-12-04
> > >
> > > We thank a reviewer for the interesting discussion.
> > >
> > > We like to note that we gave an explanation of differences between pretrained models, despite it being not the focus of the paper. We also think that questions about different pretraining strategies and the search of the threshold after which models will start working is outside of the scope this work. It will require a vast amount of compute and exploration work and will be equivalent to the design of a new set of models from scratch.
> > >
> > > In conclusion, we believe that we addressed every weakness and question from the reviewer that was possible to address in a the amount of time given during rebuttal.

---

### Author Response · Authors · 2024-11-27
**Rebuttal changes**

Dear reviewers, we appreciate your feedback and suggestions. It has allowed us to greatly improve our paper during the rebuttal period. We hope our responses will be able to address your concerns.
We shared the new revision of the paper and below are the key summary of the edits that we have made:
1. Experiments: We included many additional experiments:
   1. In the finetuning section we added the GUE Benchmark and Genomic Benchmarks as requested by the reviewer h8wc. Now combined with NT Benchmark our set of experiments for finetuning consists of 52 different tasks, and in total we ran something in the ballpark of 10,000 experiments just for finetuning.
   2. We include the feature extraction experiments on 10 different tasks from GUE as requested by the reviewer h8wc. These results are presented in Table 10 in the Appendix.
   3. We include the mutation sensitivity experiments for randomly initialized models as requested by the reviewer Q34j. These new experiments are presented in Figure 10 in the Appendix.
   4. We include the additional experiments for ClinVar sensitivity on two new genes of BRCA2 and CFTR in the main part of the paper as requested by the reviewer Q34j. These new results are reflected in the Table 3 main part of the paper.
2. Figures:
   1. We added the Teaser figure (Figure 1 in the new revision) that encapsulates three main experiment types from our paper. Consecutively we removed the old figures about the ancestry classification and mutation sensitivity experiments, since they are now incorporated in the teaser figure.
   2. We significantly changed old Figure 1 (now Figure 2) based on the feedback from the reviewer h8wc. Now instead of displaying the absolute performance it shows the difference in the performance between pretrained models and the best random model. Moreover, we added the GUE Benchmark and Genomic Benchmarks in this figure and removed the promoter and splice sites tasks from NT Benchmark since they didn’t allow to differentiate between models.
   3. Similarly we only keep the performance per subgroup for all pretrained and randomly initialized models plot only for enhancer and histone tasks on NT Benchmark (Figure 3). Similar pairwise comparison plots for NT Benchmark promoter and splice sites, GUE and Genomic Benchmarks are presented in Figure 7, Figure 8 and Figure 9 in the Appendix. For these types of plots we also added a red dashed line that indicates the performance of the best random model.
3. Text:
   1. Abstract: slightly improved wording.
   2. Introduction: improved wording and added a few citations.
   3. Models: model descriptions moved into Appendix to not duplicate the text in the Related Works section.
   4. Finetuning: big rework of the text based on the addition of the new Benchmarks. Refined Finding 1 and also added discussion about comparison of foundation models in genomics vs other fields together with Finding 2.
   5. Feature extraction: minor improvements in wording.
   6. Genomic Variation: minor improvements in wording and a few sentences to incorporate the new experiments on ClinVAR and mutation sensitivity.
   7. Related Works: major rewrite to improve wording and coherence. Added one additional citation into each subsection.
   8. Discussion: improved wording, splitted into subsections to improve the flow.
   9. Conclusion: added conclusion section.
4. Miscellaneous:
   1. Prettified Table 15: standard deviations are now shown with a smaller font.
   2. Naming of NT 50M models has been changed to NTv2 50M everywhere as we were using the v2 version throughout all of the experiments.

---

### Author Response · Authors · 2024-11-30

Dear reviewers,

We would like to kindly remind you to engage into discussion with us regarding the changes we have made in the rebuttal revision. We would greatly appreciate your thoughts on these changes and whether they address your concerns. Thank you.

---

### Note · Authors · 2025-01-24

I have read and agree with the venue's withdrawal policy on behalf of myself and my co-authors.